# Phase Transformation in UHMWPE Reactor Powders Synthesized on Various Catalysts in Mechanical and Thermal Fields

**DOI:** 10.3390/polym15040906

**Published:** 2023-02-11

**Authors:** Pavel Dorovatovskii, Marina Baidakova, Elena Ivan’kova, Vyacheslav Marikhin, Liubov Myasnikova, Roman Svetogorov, Maria Yagovkina

**Affiliations:** 1NRC “Kurchatov Institute”, pl. akad. Kurchatova 1, 123182 Moscow, Russia; 2Laboratory of Physics of Strength, Ioffe Institute of Russian Academy of Sciences, Polytekhnicheskaya 26, 194021 St. Petersburg, Russia; 3Institute of Macromolecular Compounds, Russian Academy of Sciences, V.O. Bolshoy pr. 31, 199004 St. Petersburg, Russia

**Keywords:** ultra-high molecular-weight polyethylene reactor powders, compaction, monoclinic phase, synthesis, in-situ synchrotron study

## Abstract

Nowadays, a solvent-free method for production of high performance fibers directly from ultrahigh-molecular-weight polyethylene (UHMWPE) reactor powder is being actively developed. It causes the interest in the morphology of the reactor particles and their behavior in thermal and mechanical fields. Changes in the phase composition in virgin particles of ultra-high molecular-weight polyethylene reactor powders and in particles of powders compressed at room temperature under different pressures were studied in real time using synchrotron radiation with heating in the range of 300–370 K. It was found that the content of the monoclinic phase in reactor powders depends on the type of catalyst used for synthesis and on the applied pressure. It is shown that there are monoclinic phases of different nature: a structurally stabilized monoclinic phase formed during synthesis, and a monoclinic phase resulting from plastic deformation during compaction at room temperature. The behavior of these phases in temperature and mechanical fields is compared.

## 1. Introduction

There is a significant demand for polyolefin fibers, which have found many applications, in our daily life and in a wide range of technical fields, including electrical materials, medicines, geotextiles, sports, etc. due to showing exceptional performance, excellent chemical resistance and reasonable thermal stability. Despite the large number of research papers concerned with the possible route to high performance fibers via solution spinning (so-called gel-technology) [1,2,3,4,5,6], any special progress in improving the mechanical properties of manufactured products was not observed. Currently, high performance polyethylene fibers are commercially produced by gel-technology under two trade names: Dyneema^®^ by DSM High Performance Fibers in the Netherlands and by the Toyobo/DSM joint venture in Japan. The other brand is Spectra^®^ which is produced by Honeywell in the USA. They are spun from a low concentration ultra-high molecular-weight polyethylene (UHMWPE) solution in decalin or mineral oil, making the process expensive and unsafe [3,4]. A solvent-free method for the production of high performance fibers directly from UHMWPE reactor powder (RP) was proposed by P. Smith [7,8] and commercialized by Tejin-Aramid Company that brought Endumax to the market, a high-performance UHMWPE film material. However, its strength remains slightly lower than that of gel-spun fibers, despite the fact that Tejin-Aramid uses one of the best UHMWPE reactor powders available.

To obtain strong oriented film filaments by solid-phase processing, reactor powder (RP) must first be cold-pressed, sintered at a temperature below the melting point (to turn the RP into a mechanically coherent film), and then the obtained precursor is subjected to orientation drawing (hardening). As is known, not all the UHMWPE RPs are suitable for solvent-free processing [9]. The suitability of RP for solvent-free (“dry”) processing is usually discussed in terms of the low density of “entanglements”. The entangled molecular segments localized in disordered intra-particle regions prevent the achievement of a high draw ratio and, hence, the extreme mechanical performance. As a rule, these are tie molecules of various molecular conformations passing from one structural unit to another (lamellas, shish-kebabs, nodules, etc.). The entanglement density is highly dependent on the type of catalyst system. Synthesis on single-center metallocene catalytic systems gives the best results [10,11]. The entanglement density catastrophically depends on the relationship between the polymerization rate and the crystallization rate, and is lowest when the crystallization rate is close to the polymerization one [7]. This is realized at a low synthesis temperature (about 30 °C ÷ 40 °C).

The strength of inter-particle boundaries also plays a great role in achieving high drawability. It depends on the sintering regime (temperature, pressure, and duration of sintering), in which cohesive bonds between the particles are formed. The goal of sintering is to produce a mechanically coherent monolithic precursor for further orientation drawing that will not prematurely fail before the ultimate draw ratio is reached. Sintering should be carried out at a temperature below the melting point of UHMWPE in order to maintain the original structure.

In addition, not only disordered regions, but also the type of crystal order and its behavior in a mechanical and thermal field can affect the ability of UHMWPE RP to be drawn up to high draw ratio.

As a rule, when PE crystallizes from an undisturbed melt or solution, it forms orthorhombic cells. However, the crystallization of UHMWPE in the course of slurry synthesis at a low temperature occurs under confined conditions, when a growing molecule begins to crystallize while still being attached to the catalyst particles. This cannot but affect the morphology (both crystalline and disordered regions). Indeed, the appearance of non-orthorhombic peaks was seen on the wide angle X-ray scattering (WAXS) pattern of UHMWPE RP and compacted films [12,13]. They were attributed to the metastable monoclinic phase. As is known, the monoclinic phase is stable only under stress [14,15]. The transition orthorhombic–monoclinic crystallographic cell was originally observed in a deformation experiment on PE single crystals a while ago [16].

Much attention has recently been paid to the existence of the monoclinic phase in powders, considering its presence as one of the criteria for the suitability of a powder for dry processing.

In our previous works [17], using powerful synchrotron radiation we disavowed assertions of a number of authors about the presence of a large amount of the metastable monoclinic phase (15–20%) in UHMWPE reactor powders [12,13]. Moreover, we have shown that there are two distinct monoclinic phases. One of them (M1) is a structurally stabilized phase formed during crystallization under confined conditions. The other (M2) is the result of plastic deformation, which inevitably occurs during the preparation of a monolithic sample consisting of many particles for any investigation. Reliable results can only be obtained by investigation of a single virgin particle.

The purpose of this work is a comparative study of the phase composition in the virgin single particles of UHMWPE RPs synthesized on the different catalytic systems and its transformation under heating and/or under pressure at room temperature. The change in the phase composition of the compressed films under heating is also studied.

## 2. Materials and Methods

### 2.1. Materials

Three reactor powders (RP) were selected as objects of the study (see Table 1 below).

Living aqueous polymerization using a state-of-art catalyst based on long-lived water-stable N-terphenyl salicylaldiminato Ni (II) complex leads to the formation of the nanocrystals with an unusually high degree of order that arises from the immediate deposition of the growing chain on the crystal growth front [18]. The process of aqueous polymerization is described in detail in [19,20]. It is designated as SC. The temperature and catalytic system used for synthesis of IVA-7 are shown in Table 1 while these data for Lupolen RP synthesis are unknown.

### 2.2. Methods

#### 2.2.1. Scanning Electron Microscopy

The shape and morphology of the UHMWPE RP particles were investigated using the scanning electron microscope SUPRA 55VP (Carl Zeiss, Oberkochen, Germany). The particles of nascent UHMWPE powders and/or compressed films were adhered to conducting tapes on a sample holder, coated with a layer of gold or platinum of 10–15 nm thick by sputtering in the Q150T ES (Quorum Instruments, Laughton, East Sussex, UK) and studied in the scanning microscope. The study of the samples was conducted using accelerating voltage not higher than 5 kV in order to minimize the possible distortion in the structure of the samples under investigation.

#### 2.2.2. In Situ Synchrotron Study

For the synchrotron study, the large particles (in size near 1.0 × 0.3 × 0.2 mm) from each RPs were selected. The Lupolen particles had a spherical shape, two other ones were of an irregular but different shape, as shown in Figure 1.

A single virgin particle of each RP investigated was reliably glued to the needle tip and its opposite side (the one free from the glue) was put in a beam of synchrotron X-ray radiation. The beam had a cross section of 0.4 × 0.4 mm and a wavelength λ = 0.793 Å. The comparative X-ray analysis of the single “virgin” particles of UHMWPE and of those being compressed at room temperature under various pressures (1 GPa and 5 GPa) was carried out by using synchrotron beam line Belok/XSA (NRC “Kurchatov Institute”, Moscow, Russia). The XRD measurements were performed in the transmission mode using a Rayonix SX165 area detector, which was located perpendicular to the SR beam at a distance of 80 mm. The sample was placed in a cryo-loop of 300 μm in size and accomplished a complete revolution by 360 degrees during XRD frame acquisition, which made it possible to average the diffraction patterns according to the orientations of the sample. The first one was performed at 300 K; the temperature was then raised stepwise from 300 to 373 K at a rate of about 5 K in 10 min, the temperature was kept constant during 20 min, then the particles were fast cooled to 300 K. The procedure of recording the XRD patterns is described in detail in our previous paper [15]. Conversions of the 2D XRD patterns to 1D scans were conducted with the help of the widely used Fit2d software as well as its analogues (Divis and Dionis provided by NRC “Kurchatov Institute”) and then analyzed by using the Fityk 3.0. program. To calibrate the sample–detector distance we needed a polycrystalline standard with a known position of the diffraction peaks; in this series of measurements, LaB6 (NIST SRM 660a) powder was used. The diffraction peaks were approximated by Pearson VII profile function, and the content of the M phase was calculated.

## 3. Results and Discussion

### 3.1. In Situ Wide-Angle X-ray Scattering (WAXS)

#### 3.1.1. In-Situ Wide-Angle X-ray Scattering of Individual Virgin Particles

The difference in the profiles of WAXS curves of the virgin particles synthesized on various catalysts is clearly seen in the Figure 2a and especially in Figure 2b where a smaller range of diffraction degrees is shown.

Along with 110 and 200 orthorhombic reflections, an intense monoclinic peak 001 is resolved in the WAXS pattern of the Lupolen powder, is noted in the WAXS pattern of the SC powder, and is practically undetected in the WAXS pattern of the IVA7 powder. (The difference in the intensities is caused by the various particle sizes selected for investigation).

To estimate the content of both orthorhombic and monoclinic crystalline phases in the studied particles, as well as the amount of the amorphous phase, we decomposed the experimental X-ray diffraction pattern (Figure 3) and calculated the ratio of the areas under individual peaks to the area under the experimental curve.

The content of the monoclinic phase in the separate single particles turned out to be not as high as reported in the literature [12,13] and close to our estimates made earlier for another single UHMWPE particle synthesized on an one-site metallocene catalyst [17]. It depends on the catalytic system used for the synthesis (Figure 2b). The monoclinic peak is clearly visible in the WAXS pattern of a commercial Lupolen powder, probably synthesized on a Ziegler–Natta catalytic system. It is less visible in the WAXS pattern of the SC powder. The presence of monoclinic crystals in SC particles is a surprise since synthesis in an aqueous medium leads to the formation of single crystals [18]. Its origin will be discussed below in Section 3.2. Monoclinic peak in WAXS pattern of IVA 7 powder synthesized on one-site metallocene catalyst is not detected.

It was interesting to follow the change in the phase composition after the heating of the particles from room temperature 300 K at a rate about 5 K in 10 min, keeping the temperature constant during 20 min, and the fast cooling down to 300 K. This temperature scenario leads to an increase in the content of the orthorhombic phase, mainly on the expense of the amorphous phase, i.e., the additional crystallization of a number of irregular molecular segments occurs. Monoclinic peak 001 was quite clearly seen in the diffraction pattern of an individual particle, already having disappeared at temperature 340 K and absent in the diffraction pattern after the particles were rapidly cooled to room temperature. The content of the monoclinic phase varies insignificantly, which confirms the hypothesis of the structurally stabilized origin of this phase formed during synthesis (see Figure 4 and Figure 5). The latter can only be destroyed by melting.

It should be noted that the heating–cooling cycle of the SC powder leads to the almost complete disappearance of the amorphous phase, probably, on the extent of the regularization of the folds and to the elimination of the chain tilt in some of the monoclinic crystals. Similar changes occur in the Lupolen particles, however, the content of the amorphous phase after the heating–cooling cycle remains quite high (36.8%).

#### 3.1.2. In-Situ Wide-Angle X-ray Scattering of the Compressed Virgin Particles

The appearance of X-ray diffraction patterns of pressed individual RP particles differs greatly from the X-ray diffraction patterns of the virgin particles and depends on the pressure value. The first thing that catches the eye is a significant increase in the intensity of the monoclinic peaks especially in the WAXS pattern of the Lupolen particles. It is better resolved now in the SC X-ray curve and can be resolved even in the IVA7 WAXS pattern, as shown in Figure 6a,b.

The effect of compression pressure on the appearance of WAXS patterns is demonstrated here using the RP SC particles as an example (Figure 7). X-ray diffraction patterns of the compressed particles after heating them to 370 K and rapid cooling to room temperature (rt) are also shown.

It is clearly seen that the intensity of the 001 monoclinic peak increases in particles that have been compressed at a higher pressure, while the monoclinic peaks are no longer detected in the WAXS patterns recorded for the particles that have gone through the thermal history described above.

The graphs below (Figure 8) give a quantitative idea of the change in the phase composition of particles subjected to different pressures.

A simple calculation using the numbers given in the graphs shows that in a particle compressed at a pressure of 1 GPa, the proportion of the orthorhombic crystalline phase decreases by 16% due to the destruction of 10% of orthorhombic crystallites and due to the transition of 6% of orthorhombic crystallites to monoclinic ones.

The trend towards a decrease in the crystalline phase due to an increase in the monoclinic phase and partial amorphization is even more pronounced in a particle compressed under a pressure of 5 GPa.

Heating the SC film (compressed at 5 GPa) to 370 K followed by rapid cooling to room temperature leads to an increase in the content of the orthorhombic phase and a decrease in the monoclinic and amorphous phases (Figure 9). However, the phase composition of the annealed film does not return to that before annealing.

The content of the monoclinic phase is still somewhat higher (8.5%) than in the original powder (5.84%), although the main part of the monoclinic phase, formed as a result of plastic deformation during compression, has nevertheless disappeared. This confirms our statement that the largest part of the monoclinic phase, observed by most authors, is formed mainly from orthorhombic crystallites subjected to plastic shear or tension deformation.

It is interesting to note that the change in the phase composition of the reactor powder upon compaction under pressure at room temperature occurs differently depending on the morphology of the powders themselves (see Figure 10).

In commercial Lupolen powders, whose morphology is dominated by fibrillar structural elements, no amorphization of the powder under pressure is observed. Part of the orthorhombic crystalline structure simply becomes the monoclinic.

However, on the contrary, in the loose flaky spongy-like particles of powder IVA7, the content of the amorphous component under the action of pressure decreases by more than two times, and the content of both the orthorhombic and monoclinic phases increases. Apparently, due to the large number of pores, the pressure applied to the particles lead not only to a distortion of the orthorhombic lattice in the already existing orthorhombic crystallites, but also to additional crystallization of disordered regions and the formation of orthorhombic crystallites.

The phase composition of a pressed Lupolen particle heated to 370 K and cooled to room temperature differs little from the phase composition of the original particle (Figure 11).

We have already noted above that during the synthesis a structurally stabilized monoclinic phase (M1) is formed, and another one (M2) is formed during the plastic deformation of the particles. As follows from the presented data, the largest increase in the monoclinic phase (M1 + M2) is observed for Lupolen RP particles (23.45%) compressed under a pressure of 1 GPa. The contribution of M1 to this quantity is small. SC PR particles demonstrate a smaller increase in the monoclinic phase at the same pressure (6.64%), which increases at 5 GPa to 14%, which is in any case less than that in a compressed Lupolen particle. Alas, we did not compress the IVA7 particle at 1 GPa. We only compressed it at 15 MPa, and the amount of the monoclinic phase increased by only 3%. At the same time, we observed a completely different behavior of the amorphous phase in this sample. It decreases in a compressed IVA7 particle while in the SC and Lupolen compressed particles the content of amorphous phase increases.

The question arises as to what causes the difference in the behavior of the UHMWPE reactor powders synthesized on different catalytic systems and the films compressed from these powders at room temperature in thermal and mechanical fields. To answer this question, let us turn to the results of electron microscopic studies of the structure of the powders under investigation.

### 3.2. SEM Study

The SEM micrographs of the UHMWPE powders under investigation are presented in Figure 12, Figure 13, Figure 14, Figure 15 and Figure 16. It is clear that they have quite different complicated hierarchical morphologies that can be a cause of the different response on heating and on applied pressure.

It is known that the monoclinic phase can be formed either under the action of shear stresses or under plastic tensile deformation [14,15,16,21]. The amount of the monoclinic phase formed during particle compression should depend on the presence in the particles of certain morphological formations (fibrils, lamellas, shish-kebabs) and their orientation with respect to the compressive force.

As mentioned above, it remained unclear how the monoclinic phase can be present in SC powders with a seemingly ideal structure characteristic of lamellar single crystals (Figure 12) or nanocrystals (Figure 13). At the same time, the monoclinic peak 001 was detected fairly well in the diffraction pattern of an individual SC particle (Figure 4).

It can be assumed that the monoclinic phase M1 is localized in fibrils which are rarely observed in the particles (Figure 14).

Figure 14 clearly shows the classic rearrangement of the folded structure into a fibrillar one, which is described in detail both in our [9] and in other works [6]. It is assumed that these fibrils are the secondary formations. Newly formed nanocrystals undergo plastic tensile deformation created by the growing polymer mass. It occurs in the different separate locations of the particle. The tensile force acting along the direction of the fibrils probably leads to the slippage of the molecular chain through the orthorhombic intrafibrillar crystallites. This skews them, which leads to an orthorhombic–monoclinic-M1 transition. It should be noted that the monoclinic peaks appear on the X-ray diffraction patterns of the ultimately drawn fibers when the limiting stretching is reached. It is probable that the same mechanism works.

The predominant morphological units in Lupolen RP are fibrils (Figure 15).

They are formed during synthesis in large commercial reactors on Ziegler–Natta catalysts at a temperature obviously higher than the temperature used for synthesis of the investigated lab-scale UHMWPE RP. It can be assumed that the growth of the polymer mass occurs at a higher rate, the stresses are more uniformly distributed in the polymer mass, and plastic tensile deformation leads to a multiple unfolding of lamellar crystals. This explains the appearance of a large number of fibrils.

As clearly seen in Figure 16, the morphology of IVA7 RP particles differs a lot from that of SC and Lupolen RPs.

The dominant morphological units are the well-known shish-kebabs, which consist of a central shish formed by extended chain crystals and of the kebabs formed by fold-chain lamellar crystals. Such structures are usually formed during the crystallization of polymers from low-concentration stirred solutions. Apparently, such conditions are realized during slurry synthesis in heptane in a reactor flask of small volume. The monoclinic phase M1 can be located in the central shish and in crystalline bridges (if any) between lamellar kebabs, as has been shown by using nano-focus X-ray source ID13 of the European Synchrotron Radiation Centre (Grenoble, France) for the in-situ study of individual UHMWPE RP particles synthesized on one-site metalocene catalyst [22].

## 4. Conclusions

Wide-angle X-ray diffraction using high-power synchrotron radiation was used to study the crystal structure of individual virgin particles of UHMWPE reactor powders synthesized on various catalysts and the change in thermal and mechanical fields. The obtained results confirmed the previously stated assumption about the existence of two different monoclinic phases with different origins. One of them, which we have designated M1, is formed during synthesis. Its quantity is small and is approximately 5–6%. It could only be detected using powerful synchrotron radiation. However, the M1 phase is not detected in the IVA7 powder obtained on a single-center metallocene catalyst (perhaps due to the small particle size). Since the monoclinic phase is stable only under stress, the question arises of the origin of residual stresses in the synthesis products.

One of the reasons may be the rapid growth of the polymer mass, which generates tensile forces. Tensile deformation leads to unfolding molecular folds in lamellar crystals, nanocrystals, and secondary structural units, fibrils, are formed. The transition of orthorhombic crystallites to monoclinic ones can occur only in extremely drawn fibrils in which the stress is transferred from one intra fibrillar crystal to the neighboring one by taut tie molecules. This process recalls the process of orientation hardening. The number of ultimately drawn fibrils in a polymer particle depends on the synthesis conditions: type of catalyst, synthesis temperature, stirring rate, reactor volume, ethylene feed rate, etc.

Thus, for example, in commercial Lupolen powder, in which fibrils are the dominant morphological unit, the amount of M1 should have been higher than in a powder with a single crystal structure (SC), in which there were considerably fewer fibrils. However, it is practically the same (6.1% and 5.8%, correspondingly).

Taking into account the higher synthesis temperature of the commercial Lupolen powder, one can assume that the number of strained fibrils in Lupolen powders does not much exceed the number of strained M1 fibrils in SC powder, in which there are not so many fibrils; however, due to the lower synthesis temperature, all are strained.

After heating to 370 K and rapid cooling, the amount of the monoclinic phase M1 decreases somewhat due to the partial annealing of the residual stresses. That is why we named the M1 phase as the structurally stabilized monoclinic phase. At the same time, the number of orthorhombic crystallites increases. This is realized due to the amorphous phase, the content of which is noticeably reduced. One can assume that the folds become more regular. Thus, the heating–cooling cycle leads to the formation of a more perfect structure of the reactor powder.

After applying pressure to the powder particles at room temperature, the content of the monoclinic phase (M2) increases in all particles under study. The increase in the content of the monoclinic phase is apparently caused by shear plastic deformation, which transforms orthorhombic crystals into monoclinic ones by tilting the lattice. The content of M2 depends on applied pressure and increases with pressure. It completely disappears upon heating.

Thus, the use of the content of the monoclinic phase in the powder of the reactor as a criterion for the suitability of UHMWPE RP for solid-phase processing is inappropriate. However, the results obtained are of great importance for the search for an appropriate regime for the production of precursors for the orientation drawing.

## Figures and Tables

**Figure 1 polymers-15-00906-f001:**
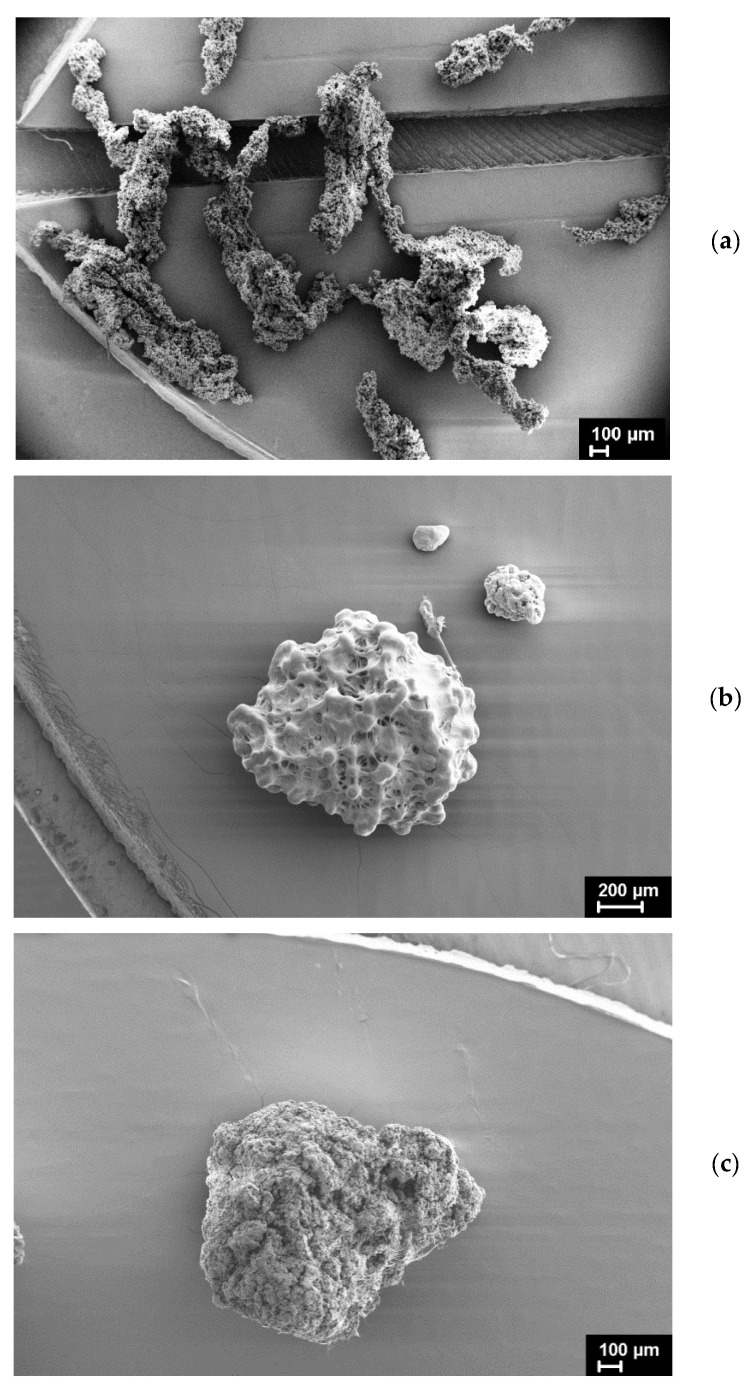
The micrographs of UHMWPE reactor powders: IVA7 (**a**), Lupolen (**b**), and SC (**c**).

**Figure 2 polymers-15-00906-f002:**
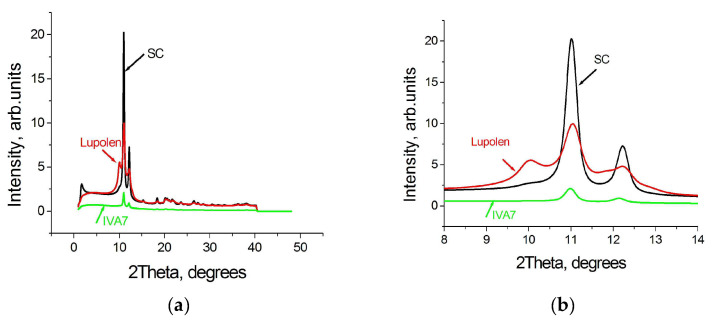
WAXS curves of the virgin individual UHMWPE RP particles: Lupolen (red), SC (black), and IVA7 (green) shown for a range 40 degrees (**a**) and 6 degrees (**b**).

**Figure 3 polymers-15-00906-f003:**
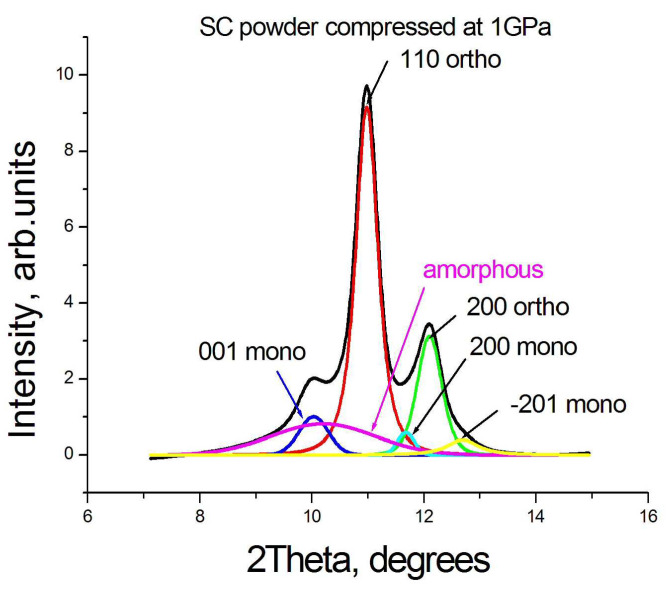
An example of the decomposition of the WAXS experimental pattern using the Fityk 3.0 program. The diffraction peaks were approximated by Pearson VII profile function.

**Figure 4 polymers-15-00906-f004:**
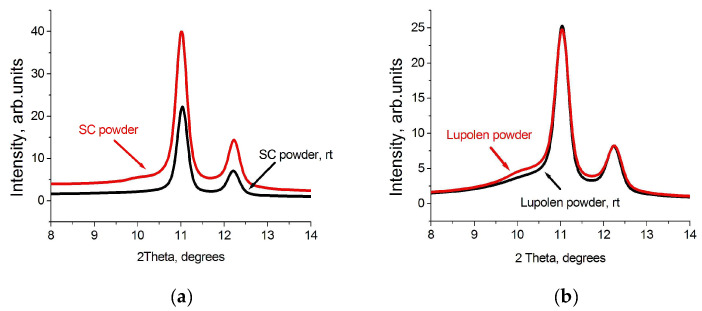
WAXS integrated patterns of SC (**a**) and Lupolen (**b**) individual particles recorded at 300 K (red) and WAXS integrated patterns recorded of the particles first heated from 300 K to 370 K with a step of 5 K, then kept 20 min at this temperature and fast cooled down to 300 K (rt, black).

**Figure 5 polymers-15-00906-f005:**
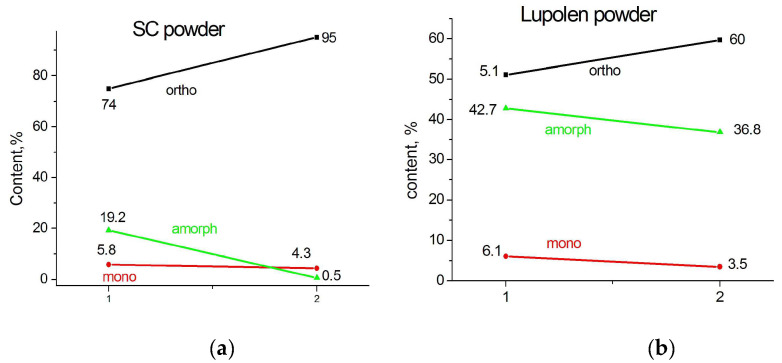
The change in phase composition of SC (**a**) and Lupolen (**b**) individual particles (1) after the heating them from 300 K up to 370 K and quickly cooling them back to 300 K (2).

**Figure 6 polymers-15-00906-f006:**
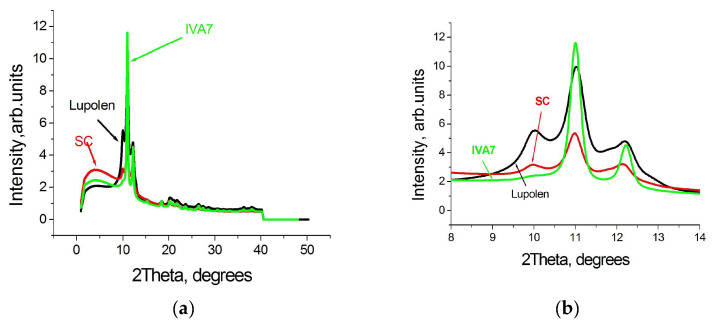
WAXS patterns of films compressed from UHMWPE RP particles Lupolen (black), SC (red) and IVA7 (green) under pressure 1 GPa at room temperature shown for a range of 50 degrees (**a**) and 6 degrees (**b**).

**Figure 7 polymers-15-00906-f007:**
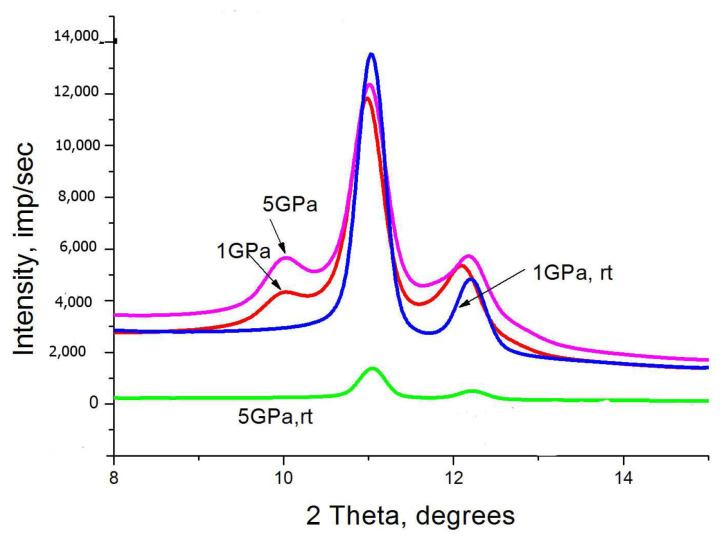
WAXS patterns of SC RP particles subjected to pressure 1 GPa and 5 GPa (shown by arrows) at room temperature. The WAXS patters designated as 1 GPa, rt and 5 GPa, rt correspond to WAXS patterns recorded at room temperature from the particles quickly cooled down to 300 K after heating.

**Figure 8 polymers-15-00906-f008:**
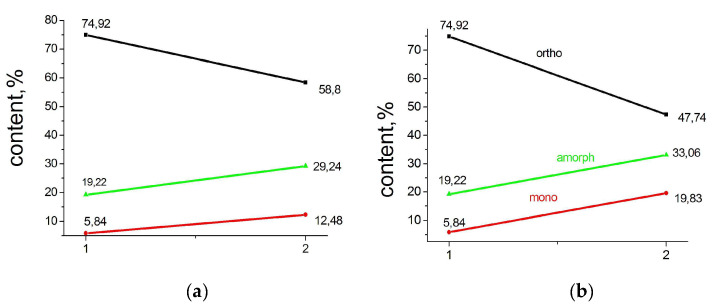
Change in the phase composition of SC particles 1 (**a**,**b**) subjected to pressure 1 GPa (2a) and 5 GPa (2b).

**Figure 9 polymers-15-00906-f009:**
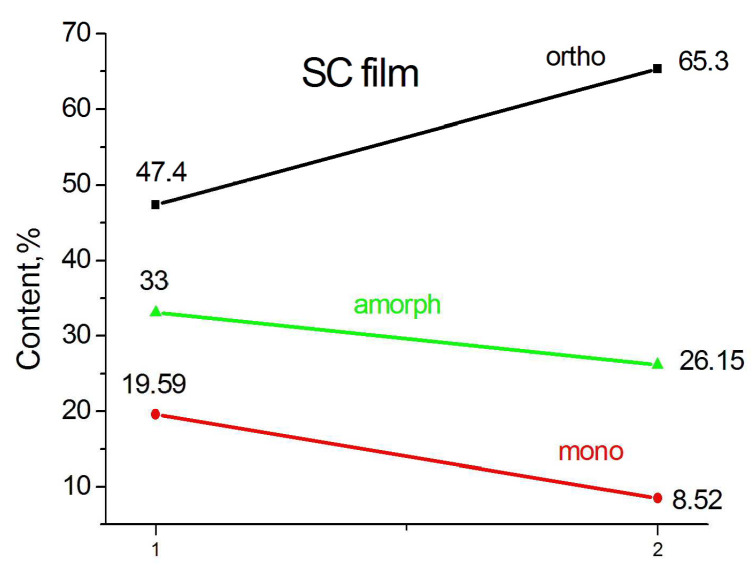
Phase composition of SC compressed film at 5 GPa (1) after its heating to 370 K and fast cooling to room temperature (2).

**Figure 10 polymers-15-00906-f010:**
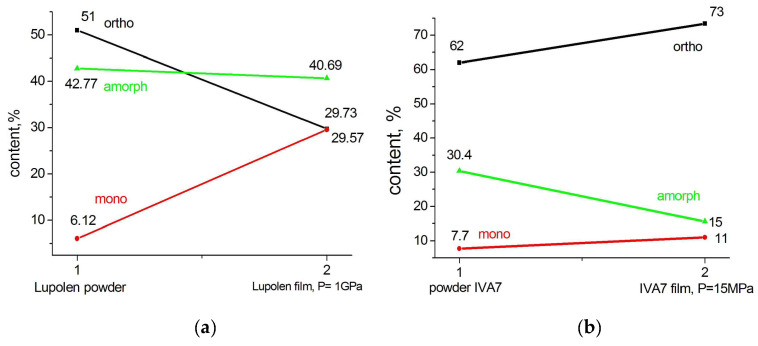
Change in phase composition of Lupolen (**a**) and IVA7 (**b**) reactor powders (1) after cold compression (2).

**Figure 11 polymers-15-00906-f011:**
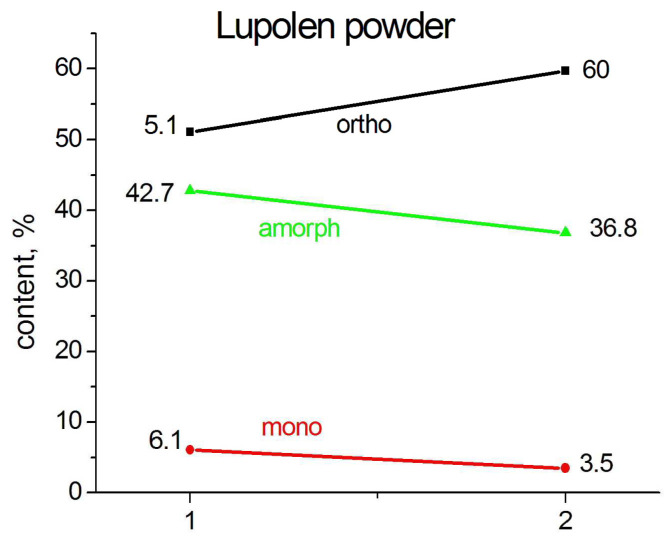
Phase composition of Lupolen film compressed at P = 1 GPa (1) after its heating to 370 K and fast cooling to room temperature (2).

**Figure 12 polymers-15-00906-f012:**
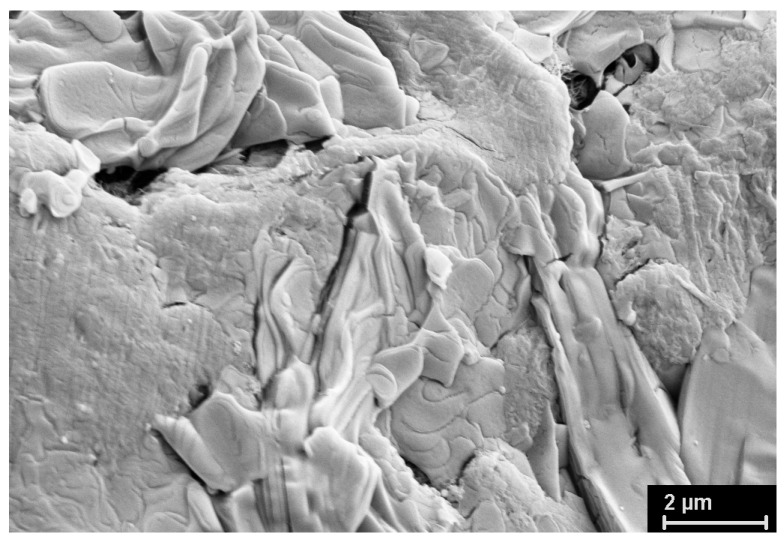
SEM micrograph of SC reactor powder.

**Figure 13 polymers-15-00906-f013:**
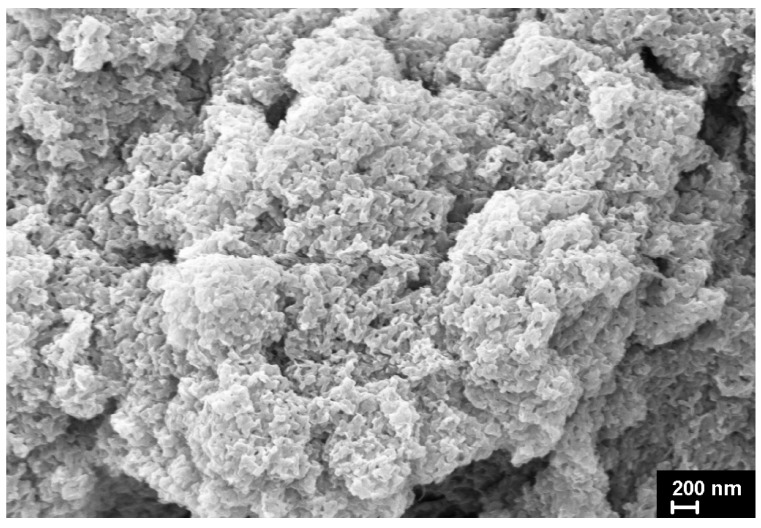
SEM micrograph of SC reactor powder (other location).

**Figure 14 polymers-15-00906-f014:**
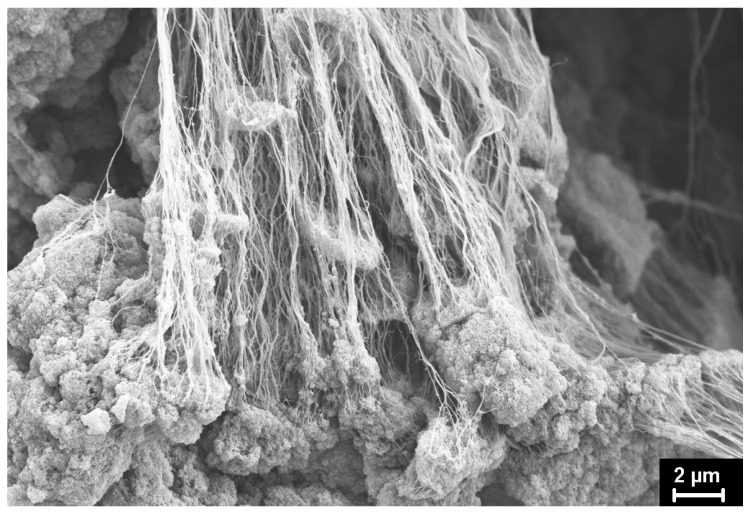
SEM micrograph of SC reactor powder (other location).

**Figure 15 polymers-15-00906-f015:**
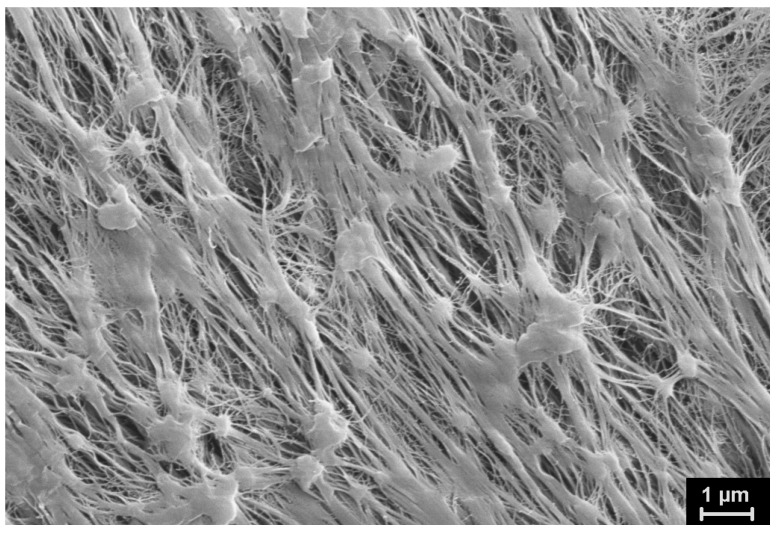
SEM micrograph of Lupolen reactor powder.

**Figure 16 polymers-15-00906-f016:**
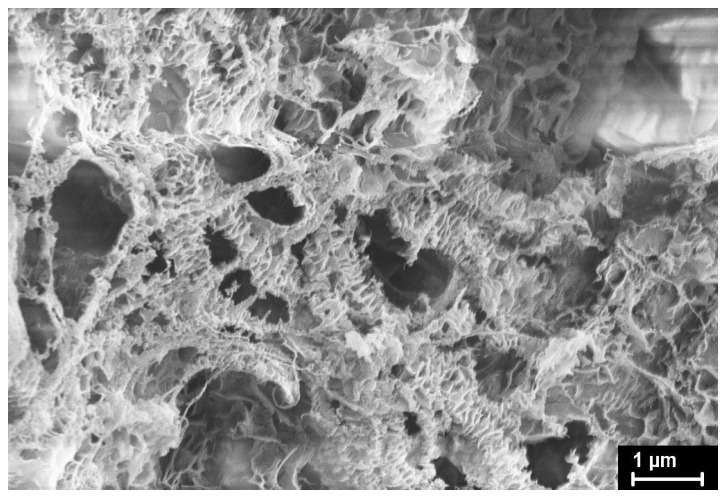
SEM micrograph of IVA7 reactor powder.

**Table 1 polymers-15-00906-t001:** Characteristics of the investigated UHMWPE reactor powders.

UHMWPE RP	Mw	T_polym_	Synthesis
SC	Mn = 4.89 × 10^6^ g/mol, Mw/Mn = 1.5);	30 °C	Living aqueous polymerization using state-of-art catalyst based on long-lived water stable complex (lab-scale) [18]
IVA-7	3.0 × 10^6^ g/mol	30 °C	One-site metallocene catalyst F-75 (lab-scale) [17]
Lupolen	5.0 × 10^6^ g/mol	-	Industrial (large-scale) reactor powder

## Data Availability

Not applicable.

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
