# Peer review of "Phase Transformation in UHMWPE Reactor Powders Synthesized on Various Catalysts in Mechanical and Thermal Fields"

_polymers, 2023, doi:10.3390/polym15040906_

Round 1

Reviewer 1 Report

The authors discuss phase transformation in UHMWPE reactor powders synthesized on various mechanical and thermal catalysts. 

1. It is preferable to use the abbreviation UHMWPE in the text (Introduction) rather than in the keyword.

2. In the material section the sentence “It was kindly provided to us by Prof. Tervoort (ETH, The 80 Switzerland) with the consent of SABIC.” can be avoided and it may be added in the acknowledgement system.

3. If the authors provide a table with the details of different synthesis, I mean the catalysts use, etc. it will give a quick idea to the readers.

4. In line 271, the reference is missing

5. It would be better the whole manuscript is checked for

            a. Language

            b. Proper citation

            c. Figure clarity

            d. Use of abbreviation

Reviewer 2 Report

In this manuscript, three different UHMWPE reactor powders were well analyzed by in-situ wide-angle X-ray scattering under different pressures with heating in the range of 300-370K. It was found that there are two different monoclinic phases. One of them (M1) is formed during synthesis under confined conditions. The other (M2) is the result of plastic deformation. Overall, I believe it can be accepted by Polymers, and I do have some minor comments for the authors to consider:

1.     The is a typo of “30÷40°Ð¡” in line 56.

2.     Why the unit of y-axis in Figure.2. is different? Can the author discuss the difference?

3.     Besides Figure 4., WAXS integrated patterns recorded from 300K with a step of 5K to 370K are requested add to the revised manuscript.

Author Response

Thank you for your hard and valuable work . I corrected the manuscript
